# The influence of prior use of inhaled corticosteroids on COVID-19 outcomes: A systematic review and meta-analysis

Chao-Hsien Chen[1,2☯], Cheng-Yi Wang[3☯], Ching-Yi Chen[4], Ya-Hui Wang[5], Kuang-Hung Chen[6], Chih-Cheng Lai[7]*, Yu-Feng Wei[8,9]*, Pin-Kuei Fu[10]*

1 Division of Pulmonary Medicine, Department of Internal Medicine, Taitung MacKay Memorial Hospital, Taitung, Taiwan, 2 Department of Medicine, MacKey Medical College, New Taipei City, Taiwan, 3 Department of Internal Medicine, Cardinal Tien Hospital and School of Medicine, College of Medicine, Fu Jen Catholic University, New Taipei City, Taiwan, 4 Division of Pulmonary Medicine, Department of Internal Medicine, E-Da Hospital, I-Shou University, Kaohsiung, Taiwan, 5 Medical Research Center, Cardinal Tien Hospital and School of Medicine, College of Medicine, Fu Jen Catholic University, New Taipei City, Taiwan, 6 Department of Internal Medicine, National Taiwan University Hospital and College of Medicine, National Taiwan University, Taipei, Taiwan, 7 Division of Hospital Medicine, Department of Internal Medicine, Chi Mei Medical Center, Tainan, Taiwan, 8 Department of Internal Medicine, E-Da Cancer Hospital, I-Shou University, Kaohsiung, Taiwan, 9 School of Medicine for International Students, College of Medicine, I-Shou University, Kaohsiung, Taiwan, 10 Department of Medical Research, Taichung Veterans General Hospital, Taichung, Taiwan

☯ These authors contributed equally to this work.
* sub164450@gmail.com (C-CL); yushuanhsu@gmail.com (Y-FW); paper199350@gmail.com (P-KF)

**Data Availability Statement:** All relevant data are within the paper and its Supporting Information files.

## Abstract

The influence of inhaled corticosteroids (ICS) on COVID-19 outcomes remains uncertain. To address this, we conducted a systematic review and meta-analysis, analyzing 30 studies, to investigate the impact of ICS on patients with COVID-19. Our study focused on various outcomes, including mortality risk, hospitalization, admission to the intensive care unit (ICU), mechanical ventilation (MV) utilization, and length of hospital stay. Additionally, we conducted a subgroup analysis to assess the effect of ICS on patients with chronic obstructive pulmonary disease (COPD) and asthma. Our findings suggest that the prior use of ICS did not lead to significant differences in mortality risk, ICU admission, hospitalization, or MV utilization between individuals who had used ICS previously and those who had not. However, in the subgroup analysis of patients with COPD, prior ICS use was associated with a lower risk of mortality compared to non-users (OR, 0.95; 95% CI, 0.90–1.00). Overall, while the use of ICS did not significantly affect COVID-19 outcomes in general, it may have beneficial effects specifically for patients with COPD. Nevertheless, more research is needed to establish a definitive conclusion on the role of ICS in COVID-19 treatment.

**PROSPERO registration number:** CRD42021279429.

## Introduction

As of March 15, 2023, the global impact of SARS-CoV-2, responsible for severe acute respiratory syndrome, has been immense, infecting over 759 million individuals and causing over 6

**Funding:** This study was funded by the Department of Medical Research of Taichung Veterans General Hospital (TCVGH-1114402D & TCVGH-1123511C) and the National Science and Technology Council (Taiwan). The funders had no role in study design, data collection and analysis, decision to publish, or preparation of the manuscript.

**Competing interests:** The authors have declared that no competing interests exist.

**Abbreviations:** SARS-COV-2, severe acute respiratory syndrome coronavirus 2; ACE2, Angiotensin-Converting Enzyme 2; COVID-19, coronavirus disease 2019; COPD, chronic obstructive pulmonary disease; CI, confidence interval; ICS, Inhaled corticosteroid; LABA, long-acting beta2-agonist; LAMA, long-acting muscarinic-antagonist; OR, odds ratio; PRISMA, Preferred Reporting Items for Systematic Reviews and Meta-analyses.

million fatalities, as reported to the World Health Organization (WHO) [1]. While vaccines and neutralizing monoclonal antibodies like tixagevimab-cilgavimab offer hope in combating COVID-19, breakthrough infections from concerning variants remain a global concern [2, 3]. Those with pre-existing conditions, such as chronic kidney disease, diabetes, hypertension, and chronic lung disease, face a higher risk of severe COVID-19 outcomes [4–6]. However, the impact of medications used to manage these comorbidities on COVID-19 outcomes remains unclear.

Inhaled corticosteroids (ICS) are commonly prescribed for conditions like asthma and chronic obstructive pulmonary disease (COPD) but may potentially increase pneumonia risk [7]. Studies during the pandemic have explored the link between ICS use and SARS-CoV-2 infections, with some suggesting potential protective effects [8–10]. A previous meta-analysis indicated that ICS treatment could improve symptom resolution and reduce hospitalization rates in mild-to-moderate COVID-19 cases [11]. However, research on the outcomes of patients with COVID-19 using ICS before infection has yielded conflicting results.

To address these discrepancies, we conducted a systematic review and meta-analysis to investigate the impact of pre-existing ICS use on outcomes of patients with COVID-19, aiming to provide clearer insights into the potential effects of ICS in this context.

## Materials and methods

This study was conducted in accordance with the Preferred Reporting Items for Systematic Reviews and Meta-Analyses (PRISMA) guidelines [12] (S1 File), and the protocol was registered under the PROSPERO registration number CRD42021279429.

### Search strategy and selection criteria

We conducted a comprehensive systematic search across multiple databases, including PubMed, Embase, Web of Science, Scopus, and the Cochrane Library, covering the period from their inception to February 28, 2023. The search terms used were "COVID-19," "SARS-CoV-2," "ICS" (including beclomethasone, budesonide, fluticasone, ciclesonide, and mometasone), "asthma," and "COPD." A detailed outline of our search strategy is provided in S1 Table. Furthermore, we manually reviewed the reference lists of relevant review articles and searched Google Scholar to identify any additional studies that met our criteria. Language restrictions were not applied. Ethical review was not applicable for this study as it does not contain any research with human participants or animals performed by any of the authors.

We included studies that met the following criteria: (1) involving adult patients with asthma, COPD, or other respiratory diseases; (2) patients had used ICS before being diagnosed with COVID-19; (3) a comparison group comprising patients who had not used ICS before COVID-19 diagnosis was available; (4) reporting on COVID-19 outcomes, such as mortality, ICU admission, hospitalization, mechanical ventilation (MV) utilization, and length of hospital stay; and (5) employing cohort or case-control designs, or randomized controlled studies. Studies lacking sufficient data for outcome analysis, as well as poster or conference abstracts, reviews or meta-analysis studies, and case reports, were excluded. Two investigators independently screened and selected each study, and any disagreements were resolved through consultation with a third investigator.

### Data extraction and outcome assessment

Data extraction was conducted independently by two investigators from all included studies, encompassing information such as publication year, study design, study population, and COVID-19 outcomes. In cases of discrepancies, discussions with a third investigator were

employed to resolve any differences. The clinical outcomes of primary interest included the assessment of mortality risk, ICU admission, hospitalization, mechanical ventilation (MV) utilization, and the duration of hospital stay.

### Risk of bias assessment

To assess the risk of bias in the included studies, three investigators independently utilized the Risk of Bias In Non-randomized Studies of Interventions (ROBINS-I) tool, evaluating 30 studies [13]. Following this, the investigators held discussions to clarify and reach a consensus on their interpretations of the questions, assessing the level of bias across seven domains and determining the overall risk for each study.

### Assessment of publication bias

Publication bias among the studies included in the meta-analyses was evaluated using a funnel plot, which illustrated the odds-ratio estimates against their standard errors. To detect any potential asymmetry in the funnel plot, statistical significance was tested using both Egger's regression intercept and Begg's rank correlation methods.

### Statistical analysis

The correlation between ICS use and dichotomous outcomes related to SARS-CoV-2 infection, such as mortality, ICU admission, hospitalization, and the use of MV, was assessed using odds ratios (ORs) with 95% confidence intervals (95% CI) [14]. When studies provided only the number and percentage of ICS use and SARS-CoV-2 infection status, ORs were computed from a two-by-two contingency table. Alternatively, ORs obtained from multivariate logistic regression models were used for analysis. An OR less than 1 indicated better outcomes for ICS users compared to non-users, suggesting a benefit for ICS users. The effect size for the length of hospital stay was represented by the standardized mean difference (SMD), where a positive SMD value indicated a longer hospital stay for ICS users.

Pooled estimates of ORs across all included studies were calculated using the DerSimonian–Laird random-effects model [15]. A significance level of two-sided $p < 0.05$ was considered if the pooled OR deviated from the null. Study heterogeneity was assessed using Cochran's Q statistic and the I2 statistic. A p-value of less than 0.10 for the Q statistic indicated statistically significant heterogeneity, while I2 values ranging from 0% to 100% represented no to extreme heterogeneity, respectively. Sensitivity analyses were performed to evaluate the impact of individual studies on the pooled estimates by employing the leave-one-out method. Subgroup analyses based on ICS use and patient characteristics were conducted. All statistical analyses were carried out using Comprehensive Meta-analysis software (version 3).

### Results

Our systematic research yielded a total of 23,983 studies, distributed across various databases as follows: 4,355 from PubMed, 11,114 from Embase, 4,698 from Web of Science, 542 from the Cochrane Library, and 3,274 from Scopus. After removing 7,437 duplicates, we further excluded 16,503 articles based on title and abstract screening. A detailed examination of the full texts resulted in the exclusion of 13 studies, leading to the inclusion of 30 studies [16–45] in our analysis (Fig 1). The characteristics of these included studies are summarized in Table 1.

Among the selected studies, all but three were retrospective analyses, with the remaining three being prospective studies [17–19]. These studies were conducted in various countries, including South Korea [16, 23, 36, 37], the US [17, 20, 22, 25, 29, 33, 34, 38, 41, 43], France

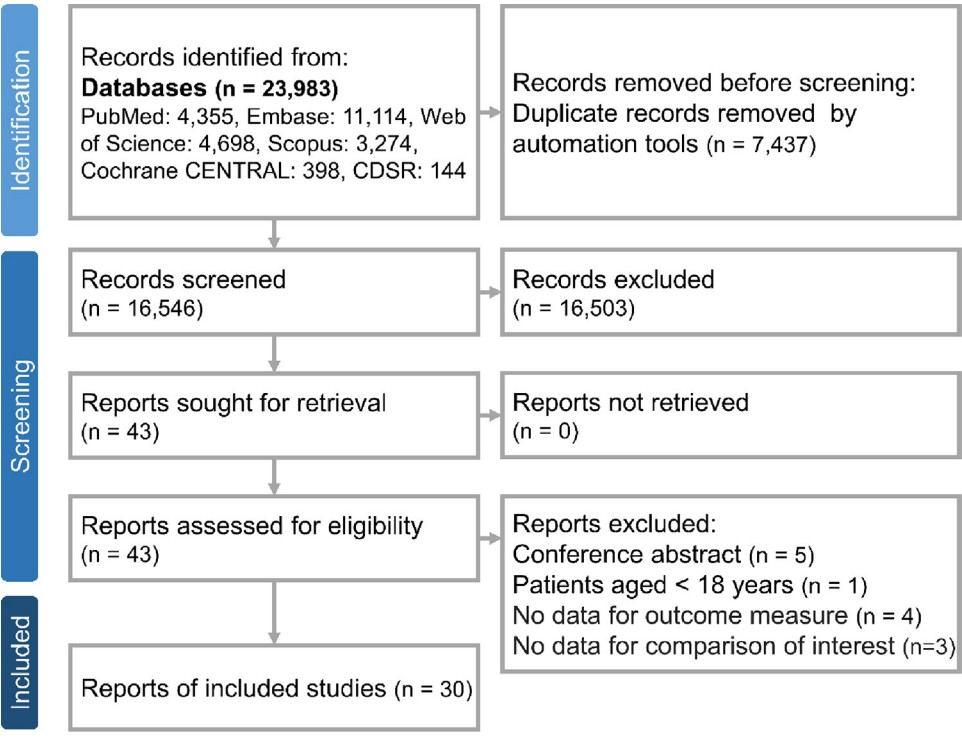

**Fig 1. The algorithm of study selection.**

[18], the UK [19, 21, 24, 42, 44], Spain [30, 31, 40, 45], Italy [26, 27], Belgium [28], China [32], Denmark [39], and Saudi Arabia [35]. Notably, all studies [16–41, 43–45] except one [42] were carried out during the initial wave of COVID-19.

## Quality assessment

The quality assessment covered seven domains and an overall risk of bias. In the domain of confounding, four studies [17, 22, 23, 25] were found to exhibit a serious risk of bias due to inadequate control for multiple co-morbidities and medications, while three studies [24, 26, 27] showed a moderate risk of bias as they adequately adjusted for multiple co-morbidities in their results. All studies demonstrated a low risk of bias in the domain concerning the selection of participants.

Regarding the classification of interventions, Argenziano et al., 2020 [17] received a serious risk of bias due to unclear information defining the intervention groups at the outset, while Graziani et al., 2020 [31] was assigned a moderate risk of bias for lacking classification details on the type and dosage of ICS used. Argenziano et al., 2020 [17] also showed a serious risk of bias in deviations from intended interventions, attributed to the recording of multiple drug usage in the study.

In the domain of bias due to missing data, 17 studies [16, 22, 23, 25–38] lacked necessary information for evaluation. Both the measurement of outcomes and selection of reported results domains were assessed as having a low risk of bias. Fig 2 provides a summary of the risk of bias for the included studies.

## Clinical outcomes

The meta-analysis results indicated that there was no significant difference in mortality risk between individuals who used ICS prior to their COVID-19 diagnosis and those who did not

**Table 1. Characteristics of the 30 included studies.**

| Source | Study design | Countries | Study period | Study subjects | No of patients |
|---|---|---|---|---|---|
| Alakeel et al., 2022 [35] | Retrospective | Saudi Arabia | March- October 2020 | Asthma with COVID-19 | 275 |
| An et al., 2021 [16] | Retrospective | South Korea | January 2019- May 2020 | Admitted for COVID-19 | 6,520 |
| Argenziano et al., 2020 [17] | Prospective | U.S.A | March- April 2020 | Admitted for COVID-19 | 1,000 |
| Aveyard et al., 2021 [24] | Retrospective | UK | January–April 2020 | COVID-19 infection | 1,090,028 (asthma); 193,520 (COPD) |
| Beurnier et al., 2020 [18] | Prospective | France | March- April 2020 | Asthma with COVID-19 | 37 |
| Bloom et al., 2021 [19] | Prospective | UK | January–August 2020 | Admitted for COVID-19 | 7,785 (asthma); 10,266 (COPD) |
| Bonato et al., 2021 [27] | Retrospective | Italy | February–November 2020 | COPD with COVID-19 | 22 |
| Calmes et al., 2021 [28] | Retrospective | Belgium | March–April 2020 | Admitted for COVID-19 | 596 |
| Chhiba et al., 2020 [34] | Retrospective | U.S.A | March- April 2020 | Asthma with COVID-19 | 220 |
| Choi et al., 2020 [23] | Nested case-control | South Korea | January–May 2020 | COVID-19 infection | 7341 |
| Choi et al., 2021 [37] | Retrospective | South Korea | January-May 2020 | Asthma with COVID-19 | 218 |
| Corradini et al., 2021 [26] | Retrospective | Italy | February–May 2020 | Admitted for COVID-19 | 3,044 |
| Dolby et al., 2023 [42] | Retrospective | UK | January 2020 –September 2021 | Patients with asthma | 2,671,931 |
| Esposito et al., 2020 [43] | Retrospective, case-control | U.S.A | March–June 2020 | ILD with COVID-19 | 46 |
| Farzan et al., 2021 [41] | Retrospective | U.S.A | March—June 2020 | Asthma with COVID-19 | 787 |
| Ferastraoaru et al., 2021 [29] | Retrospective | U.S.A | March–April 2020 | Asthma with COVID-19 | 951 |
| Graziani et al., 2020 [31] | Retrospective | Spain | January–May 2020 | COPD with COVID-19 | 793 |
| Gomez et al., 2020 [30] | Retrospective | Spain | March–May 2020 | COPD with COVID-19 | 746 |
| He et al., 2020 [32] | Retrospective | China | January–April 2020 | COPD with severe COVID-19 | 28 |
| Husby et al., 2021 [39] | Retrospective | Denmark | January–July 2020 | Admitted for COVID-19 | 6,267 |
| Inselman et al., 2021 [33] | Retrospective | U.S.A | March–June 2020 | Asthma with COVID-19 | 559,955 |
| Izquierdo et al., 2021 [40] | Retrospective | Spain | January- May 2020 | Asthma with COVID-19 | 1,006 |
| Jeong et al., 2021 [36] | Retrospective | South Korea | January–May 2020 | COVID-19 infection | 700 |
| Mather et al., 2022 [20] | Retrospective | U.S.A | February-November 2020 | Admitted for COVID-19 | 1,045 |
| Oddy et al., 2021 [44] | Retrospective | UK | January–June 2020 | Admitted for COVID-19 | 167 |
| Schultze et al., 2020 [21] | Retrospective | UK | March- May 2020 | Patients with asthma or COPD | 148,557 (COPD); 818,490 (Asthma) |
| Sen et al., 2020 [38] | Retrospective | U.S.A | March–September 2020 | COPD with COVID-19 | 1,288 |
| So et al., 2021 [22] | Retrospective | U.S.A | March–May 2020 | Admitted for COVID-19 | 408 |
| Soldevila et al., 2021 [45] | Retrospective | Spain | March–June 2020 | COVID-19 infection | 1,306 |
| Wang et al., 2020 [25] | Retrospective | U.S.A | March–June 2020 | Asthma with COVID-19 | 1,827 |

COPD, chronic obstructive pulmonary disease; ILD: interstitial lung disease; LABA, long-acting beta-2 agonist; LABD, long-acting bronchodilator

(OR, 1.01; 95% CI, 0.88–1.16; I2 = 9.8%, as depicted in Fig 3A). Moreover, the use of ICS did not significantly affect the risk of ICU admission (OR, 1.13; 95% CI, 0.78–1.63; I2 = 98.6%, Fig 3B), the risk of hospitalization (OR, 1.30; 95% CI, 0.88–1.92; I2 = 98.6%, Fig 3C), or the need for mechanical ventilation (OR, 1.02; 95% CI, 0.76–1.37; I2 = 19.3%, Fig 3D), compared to those who did not use ICS. Additionally, there was no significant difference in the length of hospital stay between ICS users and non-ICS users (SMD, 0.05; 95% CI, -0.04 to 0.14; I2 = 0%, Fig 3E). These findings remained consistent in the leave-one-out sensitivity analysis (S1A–S1E Fig), indicating the robustness of the results across various scenarios.

(A)

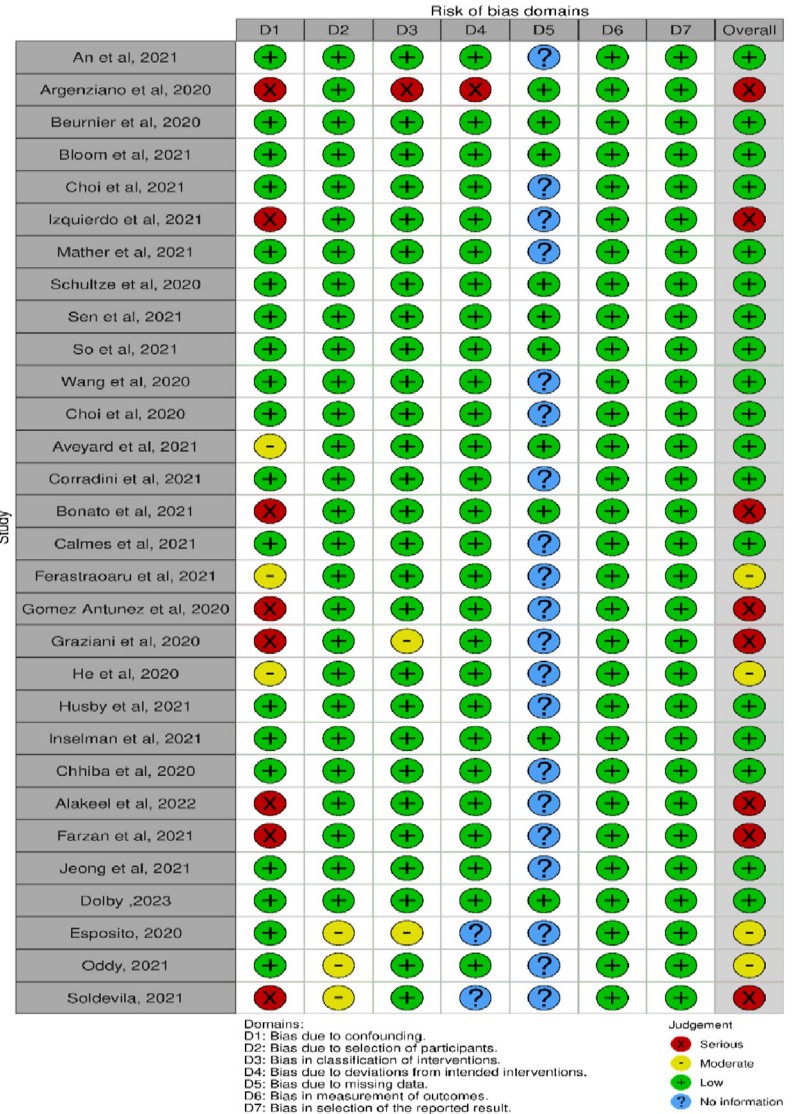

(B)

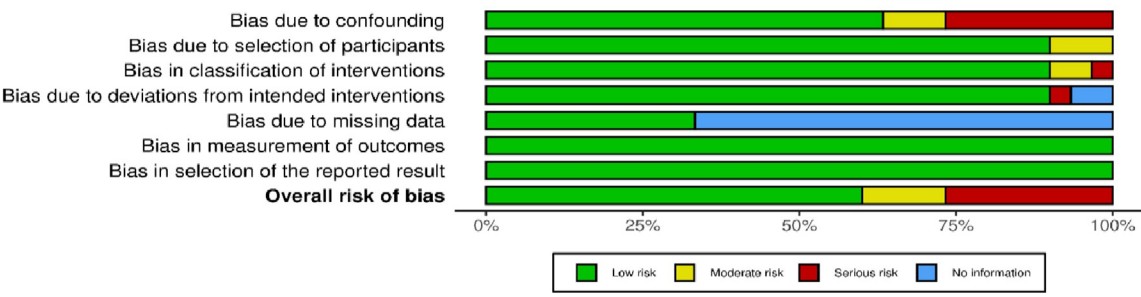

**Fig 2.** The risk of bias for the included studies (A), and summarized figure (B).

(A) Mortaility

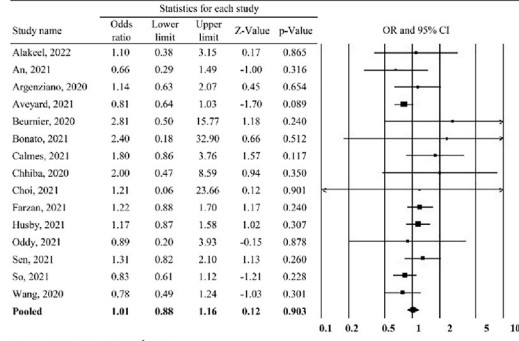

(B) ICU admission

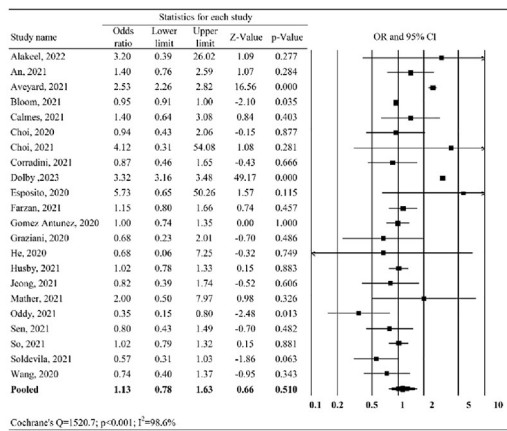

(C) Hospitalization

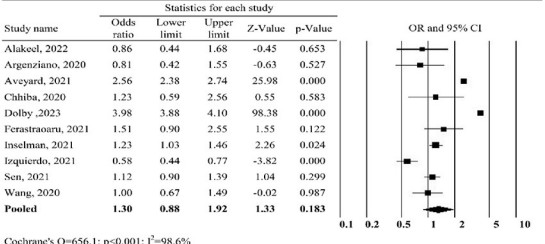

(D) Mechanical ventilation use Length of hospital stay

| Study name | Odds ratio | Lower limit | Upper limit | Z-Value | p-Value | OR and 95% CI |
|---|---|---|---|---|---|---|
| Alakeel, 2022 | 1.50 | 0.48 | 4.66 | 0.69 | 0.487 | |
| Choi, 2020 | 0.92 | 0.22 | 3.77 | -0.12 | 0.906 | |
| Farzan, 2021 | 1.17 | 0.81 | 1.69 | 0.83 | 0.406 | |
| Sen, 2021 | 1.65 | 0.68 | 3.98 | 1.11 | 0.265 | |
| So, 2021 | 0.74 | 0.51 | 1.07 | -1.59 | 0.111 | |
| **Pooled** | **1.02** | **0.76** | **1.37** | **0.11** | **0.912** | |

Cochrane's Q=5.0; p=0.292; I²=19.3%

(E) Length of hospital stay

| Study name | Std diff in means | Lower limit | Upper limit | Z-Value | p-Value | Std diff in means and 95% CI |
|---|---|---|---|---|---|---|
| Alakeel, 2022 | -0.05 | -0.31 | 0.21 | -0.36 | 0.716 | |
| An, 2021 | 0.02 | -0.12 | 0.17 | 0.29 | 0.770 | |
| Choi, 2021 | -0.06 | -0.34 | 0.23 | -0.39 | 0.696 | |
| Farzan, 2021 | 0.13 | -0.02 | 0.27 | 1.73 | 0.083 | |
| **Pooled** | **0.05** | **-0.04** | **0.14** | **1.02** | **0.308** | |

Cochrane's Q=2.3; p=0.506; I²=0%

**Fig 3.** Forest plot of the risk of mortality (A), ICU admission (B), hospitalization (C), mechanical ventilation use (D) and length of hospital stay (E) between prior inhaled corticosteroid (ICS) use and non-ICS use.

## Subgroup analysis

The subgroup analysis did not reveal a significant difference between prior ICS users and non-ICS users with asthma in terms of the risk of mortality (OR, 1.45; 95% CI, 0.74–2.84), ICU admission (OR, 1.06; 95% CI, 0.83–1.36), hospitalization (OR, 1.25; 95% CI, 0.59–2.63), and mechanical ventilation use (OR, 1.19; 95% CI, 0.84–1.69) (Table 2).

When comparing ICS alone therapy to non-ICS use, similar risks of mortality (OR, 1.56; 95% CI, 0.71–3.41), ICU admission (OR, 0.97; 95% CI, 0.74–1.27), and hospitalization (OR, 1.41; 95% CI, 0.61–3.27) were observed. Similarly, comparable risks of mortality (OR, 1.35; 95% CI, 0.58–3.16), ICU admission (OR, 0.82; 95% CI, 0.57–1.17), and hospitalization (OR, 1.91; 95% CI, 0.86–4.24) were found when comparing ICS plus long-acting bronchodilator (LABD) use to non-ICS use. Lastly, no significant difference was noted in mortality (OR, 1.11; 95% CI, 0.58–32.12) and ICU admission (OR, 1.05; 95% CI, 0.65–1.69) when comparing ICS use with LABD use (Table 2).

The prospective and retrospective studies were analyzed separately. In the prospective studies, prior ICS users showed a marginally lower risk of mortality (OR, 0.95; 95% CI, 0.91–1.00) and a similar risk of ICU admission (OR, 2.68; 95% CI, 0.64–11.31) compared to non-ICS users. In the subgroup analysis of retrospective studies, there was no significant difference in

**Table 2. Subgroup analyses of the risk of ICS use on study outcomes.**

| Outcome | Subgroup | No. of studies | Statistics for each study | | | | | Heterogeneity | | |
|---|---|---|---|---|---|---|---|---|---|---|
| | | | Odds ratio | Lower limit | Upper limit | Z-Value | p-Value | Q | p-value | I² |
| Mortality | Asthma | 9 | 1.45 | 0.74 | 2.84 | 1.08 | 0.280 | 357.7 | <0.001 | 97.8% |
| | COPD | 8 | 0.95 | 0.90 | 1.00 | -2.16 | 0.031 | 2.7 | 0.911 | 0.0% |
| | ICS alone vs. no ICS | 3 | 1.56 | 0.71 | 3.41 | 1.12 | 0.265 | 71.9 | <0.001 | 97.2% |
| | ICS/LABA vs. no ICS | 4 | 1.35 | 0.58 | 3.16 | 0.70 | 0.483 | 248.6 | <0.001 | 98.8% |
| | ICS vs. no ICS | 22 | 1.13 | 0.78 | 1.63 | 0.66 | 0.510 | 1520.7 | <0.001 | 98.6% |
| | ICS vs. others | 2 | 1.11 | 0.58 | 2.12 | 0.30 | 0.762 | 11.3 | 0.001 | 91.2% |
| | Prospective study | 1 | 0.95 | 0.91 | 1.00 | -2.10 | 0.035 | NAᵃ | | |
| | Retrospective study | 21 | 1.14 | 0.83 | 1.55 | 0.82 | 0.411 | 367.1 | <0.001 | 94.6% |
| ICU admission | Asthma | 7 | 1.06 | 0.83 | 1.36 | 0.48 | 0.631 | 5.8 | 0.449 | 0.0% |
| | COPD | 2 | 1.34 | 0.84 | 2.12 | 1.22 | 0.221 | 0.2 | 0.655 | 0.0% |
| | ICS alone vs. no ICS | 3 | 0.97 | 0.74 | 1.27 | -0.23 | 0.816 | 1.1 | 0.565 | 0.0% |
| | ICS/LABA vs. no ICS | 4 | 0.82 | 0.57 | 1.17 | -1.09 | 0.277 | 5.0 | 0.173 | 39.8% |
| | ICS vs. no ICS | 15 | 1.01 | 0.88 | 1.16 | 0.12 | 0.903 | 15.5 | 0.344 | 9.8% |
| | ICS vs. others | 2 | 1.05 | 0.65 | 1.69 | 0.21 | 0.832 | 0.3 | 0.587 | 0.0% |
| | Prospective study | 2 | 2.68 | 0.64 | 11.31 | 1.34 | 0.179 | 0.0 | 0.921 | 0.0% |
| | Retrospective study | 13 | 1.00 | 0.87 | 1.15 | 0.01 | 0.993 | 13.7 | 0.322 | 12.2% |
| Hospitalization | Asthma | 7 | 1.25 | 0.59 | 2.63 | 0.59 | 0.558 | 425.3 | <0.001 | 98.6% |
| | COPD | 1 | 1.12 | 0.90 | 1.39 | 1.04 | 0.299 | NAᵃ | | |
| | ICS alone vs. no ICS | 2 | 1.41 | 0.61 | 3.27 | 0.81 | 0.420 | 4.0 | 0.045 | 75.1% |
| | ICS/LABA vs. no ICS | 3 | 1.91 | 0.86 | 4.24 | 1.59 | 0.112 | 34.9 | <0.001 | 94.3% |
| | Prospective study | 0 | NAᵃ | | | | | NAᵃ | | |
| | Retrospective study | 10 | 1.30 | 0.88 | 1.92 | 1.33 | 0.183 | 656.1 | <0.001 | 98.6% |
| Ventilation | Asthma | 3 | 1.19 | 0.84 | 1.69 | 0.99 | 0.321 | 0.2 | 0.909 | 0.0% |
| | COPD | 2 | 1.38 | 0.61 | 3.10 | 0.78 | 0.435 | 1.0 | 0.318 | 0.0% |
| | Prospective study | 0 | NAᵃ | | | | | NAᵃ | | |
| | Retrospective study | 5 | 1.02 | 0.76 | 1.37 | 0.11 | 0.912 | 5.0 | 0.292 | 19.3% |

ᵃOnly one or none of study was included. COPD, chronic obstructive pulmonary disease; ICS: inhaled corticosteroid; ICU: intensive care unit; LABA, long-acting beta-2 agonist; NA, not applicable

the risk of mortality (OR, 1.14; 95% CI, 0.83–1.55), ICU admission (OR, 1.00; 95% CI, 0.87–1.15), hospitalization (OR, 1.30; 95% CI, 0.88–1.92), or mechanical ventilation use (OR, 1.02; 95% CI, 0.76–1.37).

## Publication bias

Funnel plots for mortality (Fig 4A), ICU admission (Fig 4B), hospitalization (Fig 4C), mechanical ventilation use (Fig 4D), and length of hospital stay (Fig 4E) were presented. Publication bias was observed for hospitalization (Egger's test: t = 4.35, p = 0.002) and mortality (Begg's test: z = 2.7, p = 0.005) (Table 3).

## Discussion

In our comprehensive meta-analysis of 30 studies, we found that the previous use of inhaled corticosteroids (ICSs) did not significantly affect COVID-19-related outcomes, including mortality, ICU admission, hospitalization, use of mechanical ventilation, or the length of hospital stay. These findings were consistent across our leave-one-out sensitivity analysis and in our

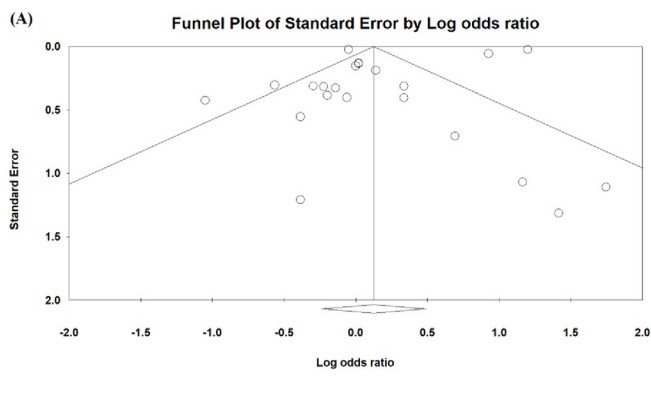

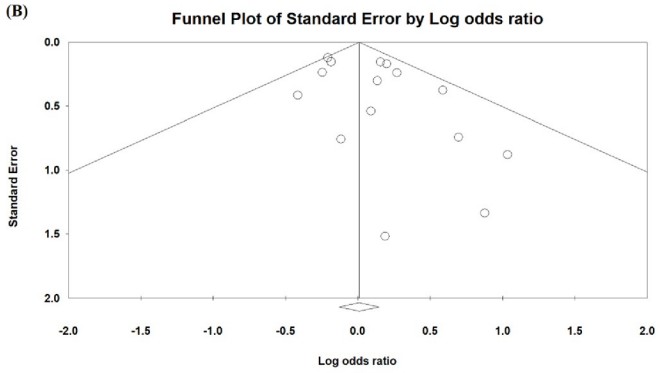

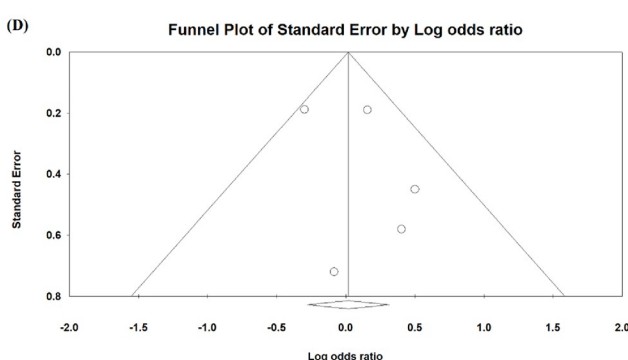

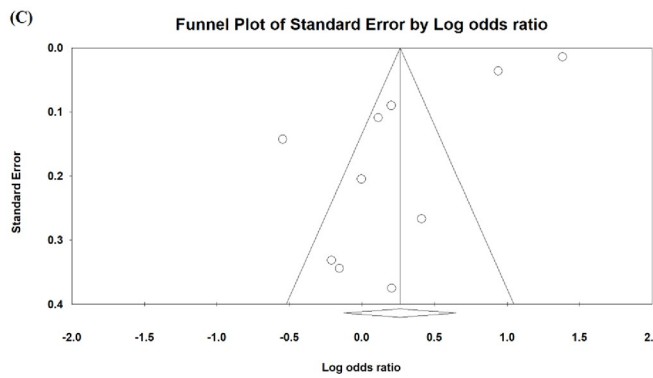

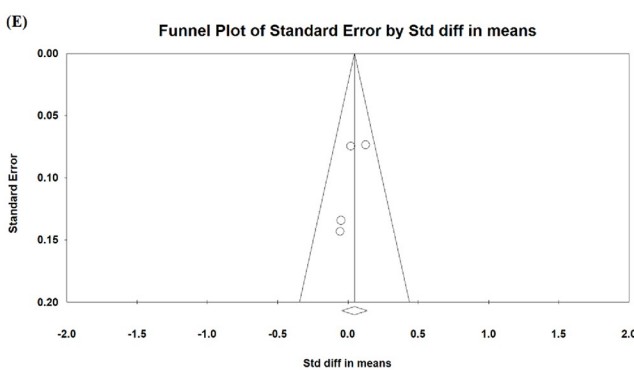

**Fig 4.** Funnel plot of mortality (A), ICU admission (B), hospitalization (C), mechanical ventilation use (D) and length of hospital stay (E).

**Table 3. Testing for symmetry of funnel plots for the risk of ICS use on study outcomes.**

| Outcomes | Egger's regression intercept method | | | | Begg's rank correlation | | |
|---|---|---|---|---|---|---|---|
| | Intercept (SE) | t-value | df | p-value | τ | z-value | p-value |
| Mortality | -1.52 (2.18) | 0.70 | 20 | 0.494 | 0.43 | 2.79 | 0.005 |
| ICU admission | 0.74 (0.45) | 1.64 | 13 | 0.125 | 0.14 | 0.74 | 0.458 |
| Hospitalization | -8.32 (1.91) | 4.35 | 8 | 0.002 | 0.29 | 1.16 | 0.245 |
| Mechanical ventilation | 0.93 (1.07) | 0.87 | 3 | 0.450 | 0.00 | 0.00 | 0.999 |
| Length of hospital stay | -2.00 (1.20) | 1.67 | 2 | 0.237 | -0.67 | 1.36 | 0.174 |

SE, standard error; df, degree of freedom

subgroup analysis of patients with asthma or COPD. Notably, our subgroup analysis revealed that patients with COPD who had previously used ICSs had a slightly lower mortality risk compared to non-users.

Furthermore, our subgroup analysis, comparing ICS therapy with long-acting bronchodilator (LABD) therapy in patients stratified by ICS regimen (with or without LABD therapy), confirmed our overall results. These results align with a previous meta-analysis by Kow and Hasan, which also concluded that pre-admission use of ICS did not increase the risk of severe or fatal COVID-19 [13]. Overall, our meta-analysis suggests that prior use of ICS did not significantly influence COVID-19-related outcomes, affirming the safety of ICS use during the pandemic. These findings are consistent with recommendations from the Global Initiative for Asthma (GINA) and the Global Initiative for Chronic Obstructive Lung Disease (GOLD) regarding ICS use in patients with asthma or COPD [46, 47].

A previous large-scale observational cohort study conducted in the United Kingdom suggested that prescribing inhaled corticosteroids (ICSs) to patients with chronic obstructive pulmonary disease (COPD) and high-dose ICSs to patients with asthma were associated with an increased risk of COVID-19-related death [21]. However, another UK national prospective cohort study reported that the use of ICSs within 2 weeks of admission may improve survival for patients with asthma aged 50 years or older, but not for patients with COPD [19]. Despite these findings, several other studies have demonstrated that ICSs neither benefited nor adversely affected COVID-19-related outcomes. A nationwide cohort study in Korea found no correlation between asthma-related medication or asthma severity and the clinical outcomes of COVID-19 after adjusting for confounding factors [37]. Additionally, several retrospective studies have reported that ICSs did not influence COVID-19-related outcomes [20, 39, 41]. Similarly, our current meta-analysis of 26 studies indicated that prior use of ICSs before SARS-CoV-2 infection did not impact COVID-19-related outcomes. This conclusion remained consistent after conducting a leave-one-out sensitivity test.

Our subgroup analysis revealed that prior ICS use was associated with a reduced risk of mortality in patients with COPD, but not in those with asthma. One possible explanation for this discrepancy might be related to chronic type 2 airway inflammation induced by interleukin (IL)-13 in the airway epithelium, which could elevate ADAM17 expression, leading to the downregulation of ACE2 expression before COVID-19 infection in patients with asthma [48]. The interaction between ICS treatment and gene expression in patients with COPD is intricate. A previous study indicated that ICS treatment diminished ADAM17 expression in patients with COPD, irrespective of their smoking status at baseline, but it did not impact ACE2 [8]. A combination of formoterol/budesonide and salmeterol/fluticasone ICS treatments seemed to downregulate genes associated with ACE2 and ADAM17/FURIN [8]. However, the OpenSAFELY study suggested that ICS use neither demonstrably reduces nor increases COVID-19-related mortality in people with COPD [21]. Given the lack of epidemiological evidence suggesting that ICS therapy escalates COVID-19 severity or mortality, we align with the international consensus that ICS therapy should be continued in patients with COPD if clinically indicated until further evidence becomes available.

This meta-analysis has several limitations. Firstly, significant heterogeneity was observed in some of our findings, which could be attributed to variations in study designs. Despite conducting subgroup analyses to address this heterogeneity, it remained in some cases. Due to insufficient data, we could not assess the impact of various study designs, asthma phenotypes, timing, duration, dosage intensity, or types of ICS on COVID-19-related outcomes. Furthermore, the effect size relative to ICS dosage was not evaluable, as only four studies [18, 21, 29, 42] reported different ICS dosages, each with varying outcomes. Additional research is needed to address this issue. Thirdly, the concurrent use of LABD with ICS could potentially confound

our results. Nonetheless, our findings were consistent even when the analysis was stratified by the use of ICS alone or in combination with LABD therapy, as compared to the use of no ICS. Moreover, the leave-one-out sensitivity analysis confirmed the stability of these findings, thereby reinforcing the credibility of our results. Finally, we did not examine the potential confounding influence of continuing ICS therapy after contracting the SARS-CoV-2 infection. Further research is required to delve into this matter.

## Conclusions

Our meta-analysis did not find a significant link between prior use of ICS and COVID-19 outcomes, including mortality, ICU admission, hospitalization, mechanical ventilation use, and length of hospital stay. However, interestingly, our results suggest that prior use of ICS may be beneficial for COVID-19 outcomes in patients with COPD. These findings are in line with the recommendations from GINA and GOLD, which support the use of ICSs for patients with chronic respiratory diseases. Overall, our study provides important insights into the safety and potential benefits of ICS use in the context of the COVID-19 pandemic, particularly for patients with COPD. Further research is warranted to explore these findings in more detail and to guide clinical management effectively.

## Supporting information

**S1 Fig.** Leave-one-out sensitivity test of the risk of mortality (A), ICU admission (B), hospitalization (C), mechanical ventilation use (D) and length of hospital stay (E) between inhaled corticosteroid (ICS) use and non-use.
(DOCX)

**S1 Table. Search strategy.**
(DOCX)

**S1 File. PRISMA checklist.**
(DOCX)

## Author Contributions

**Conceptualization:** Chao-Hsien Chen, Ching-Yi Chen, Chih-Cheng Lai.

**Data curation:** Cheng-Yi Wang, Ching-Yi Chen, Ya-Hui Wang, Kuang-Hung Chen.

**Formal analysis:** Chao-Hsien Chen, Ching-Yi Chen, Ya-Hui Wang, Kuang-Hung Chen, Yu-Feng Wei, Pin-Kuei Fu.

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
