## [Decision Letter · Decision Letter 0]

17 Oct 2023

PONE-D-23-28841The Influence of Prior Use of Inhaled Corticosteroids on COVID-19 Outcomes: A Systematic Review and Meta-AnalysisPLOS ONE

Dear Dr. Fu,

Thank you for submitting your manuscript to PLOS ONE. After careful consideration, we feel that it has merit but does not fully meet PLOS ONE’s publication criteria as it currently stands. Therefore, we invite you to submit a revised version of the manuscript that addresses the points raised during the review process.

We look forward to receiving your revised manuscript.

Kind regards,

Dong Keon Yon, MD, FACAAI, FAAAAI

Academic Editor

PLOS ONE

Journal Requirements:

 "The authors sincerely appreciate funding in part by the Department of Medical Research of Taichung Veterans General Hospital (TCVGH-1114402D & TCVGH-1123511C) and the National Science and Technology Council (Taiwan) (NSTC 112-2314-B-075A-003 -MY3) for the supporting of study manpower, materials, and the publication fees."

7. We notice that your supplementary figures are uploaded with the file type 'Figure'. Please amend the file type to 'Supporting Information'. Please ensure that each Supporting Information file has a legend listed in the manuscript after the references list.

8. We notice that your supplementary tables are included in the manuscript file. Please remove them and upload them with the file type 'Supporting Information'. Please ensure that each Supporting Information file has a legend listed in the manuscript after the references list.

**Additional Editor Comments:**

Thank you for submitting your manuscript. The reviewers and I believe it is of potential value for our readers. However, the reviewers have raised a number of very important issues, and their excellent comments will need to be adequately addressed in a revision before the acceptability of your manuscript for publication in the Journal can be determined. We cannot guarantee that your revised paper will be chosen for publication; this would be solely based on how satisfactorily you have addressed the reviewer comments.

#1. COVID-19 patients. -> patients with COVID-19

Please use the patient-first language.

#2.A significance level of p<0.05 -> A significance level of two-sided p<0.05

#3. was assessed using odds ratios (ORs) with 95% confidence intervals (95% CI). -> Please cite the statistical guideline (DOI: https://doi.org/10.54724/lc.2022.e9).

Reviewers' comments:

Reviewer's Responses to Questions

**Comments to the Author**

1. Is the manuscript technically sound, and do the data support the conclusions?

Reviewer #1: Yes

Reviewer #2: Yes

2. Has the statistical analysis been performed appropriately and rigorously? 

Reviewer #1: Yes

Reviewer #2: Yes

3. Have the authors made all data underlying the findings in their manuscript fully available?

Reviewer #1: Yes

Reviewer #2: Yes

4. Is the manuscript presented in an intelligible fashion and written in standard English?

Reviewer #1: Yes

Reviewer #2: Yes

5. Review Comments to the Author

Reviewer #1: Dear authors I congratulate to you for your outstanding work and research.

1. The introduction provides clear context and rationale for the study, referencing relevant research, and citing sources appropriately. The safety of ICD in COVID pandemic has been severly debated with more or less weak evidence.

2. The inclusion of PROSPERO registration number adds transparency to the research process in Methods.

3. The process of data extraction and outcome assessment is described adequately, including details on how discrepancies were resolved among investigators.

4. The use of the ROBINS-I tool for assessing the risk of bias in the included studies is appropriate, and the results are presented clearly.

5. The assessment of publication bias is appropriately conducted using funnel plots and statistical tests.

6. The use of sensitivity analyses to test the robustness of the results is a good practice.

7. Figures and tables are used effectively to present the findings, making them easily comprehensible.

8. The discussion section provides a comprehensive interpretation of the results.

9. While the paper acknowledges limitations, it would be beneficial to discuss them in more detail, particularly regarding study heterogeneity and potential confounders such as dosage of inhaled corticosteroids which are not disclosed in the analysis. It may be that we would see some signal of lower hospital admission in high dose ICS group. You mention that "results remained consistent even after stratifying the analysis by ICS monotherapy or ICS in combination with LABD therapy" which should not have any effect on the hypothesis or does it explain the confounding factors.

Overall, this scientific paper is well-structured and provides a thorough investigation into the impact of pre-existing ICS use on COVID-19 outcomes. It adheres to established guidelines and offers valuable insights into a relevant clinical question.

Reviewer #2: Dear Authors,

The research topic, which examines the effects of previous use of inhaled corticosteroids on the outcomes of COVID-19, is not a new subject. However, the systematic review and meta-analysis has been conducted with acceptable precision and well documented. The only recommendation is to include the number of studies used in each subgroup analysis within the text.

6. PLOS authors have the option to publish the peer review history of their article (what does this mean?). If published, this will include your full peer review and any attached files.

Reviewer #1: No

Reviewer #2: No

---

## [Author Response · Author response to Decision Letter 0]

8 Nov 2023

Editor Comments:

Thank you for submitting your manuscript. The reviewers and I believe it is of potential value for our readers. However, the reviewers have raised a number of very important issues, and their excellent comments will need to be adequately addressed in a revision before the acceptability of your manuscript for publication in the Journal can be determined. We cannot guarantee that your revised paper will be chosen for publication; this would be solely based on how satisfactorily you have addressed the reviewer comments.

#1. COVID-19 patients. -> patients with COVID-19

Please use the patient-first language.

Reply: Thank you for your suggestion. We have made the corrections accordingly.

#2. A significance level of p<0.05 -> A significance level of two-sided p<0.05

Reply: Thank you for your comment. We have made the correction based on your suggestion.

#3. was assessed using odds ratios (ORs) with 95% confidence intervals (95% CI). -> Please cite the statistical guideline (DOI: https://doi.org/10.54724/lc.2022.e9).

Reply: Thank you for your comment. We have cited the reference according to your suggestion.

Reviewers Comments:

Reviewer #1: Dear authors I congratulate to you for your outstanding work and research.

1. The introduction provides clear context and rationale for the study, referencing relevant research, and citing sources appropriately. The safety of ICD in COVID pandemic has been severely debated with more or less weak evidence.

Reply: Thank you for your comment.

2. The inclusion of PROSPERO registration number adds transparency to the research process in Methods.

Reply: Thank you for your comment.

3. The process of data extraction and outcome assessment is described adequately, including details on how discrepancies were resolved among investigators.

Reply: Thank you for your comment.

4. The use of the ROBINS-I tool for assessing the risk of bias in the included studies is appropriate, and the results are presented clearly.

Reply: Thank you for your comment.

5. The assessment of publication bias is appropriately conducted using funnel plots and statistical tests.

Reply: Thank you for your comment.

6. The use of sensitivity analyses to test the robustness of the results is a good practice.

Reply: Thank you for your comment.

7. Figures and tables are used effectively to present the findings, making them easily comprehensible.

Reply: Thank you for your comment.

8. The discussion section provides a comprehensive interpretation of the results.

Reply: Thank you for your comment.

9. While the paper acknowledges limitations, it would be beneficial to discuss them in more detail, particularly regarding study heterogeneity and potential confounders such as dosage of inhaled corticosteroids which are not disclosed in the analysis. It may be that we would see some signal of lower hospital admission in high dose ICS group. You mention that "results remained consistent even after stratifying the analysis by ICS monotherapy or ICS in combination with LABD therapy" which should not have any effect on the hypothesis or does it explain the confounding factors.

Reply: Thank you for your comment. We have revised the limitations section to emphasize the issues of study heterogeneity, ICS dosage, and concurrent use of LABD. “This meta-analysis has several limitations. Firstly, significant heterogeneity was observed in some of our findings, which could be attributed to variations in study designs. Despite conducting subgroup analyses to address this heterogeneity, it remained in some cases. Due to insufficient data, we could not assess the impact of various study designs, asthma phenotypes, timing, duration, dosage intensity, or types of ICS on COVID-19-related outcomes. Furthermore, the effect size relative to ICS dosage was not evaluable, as only four studies [18, 21, 29, 42] reported different ICS dosages, each with varying outcomes. Additional research is needed to address this issue. Thirdly, the concurrent use of LABD with ICS could potentially confound our results. Nonetheless, our findings were consistent even when the analysis was stratified by the use of ICS alone or in combination with LABD therapy, as compared to the use of no ICS. Moreover, the leave-one-out sensitivity analysis confirmed the stability of these findings, thereby reinforcing the credibility of our results. Finally, we did not examine the potential confounding influence of continuing ICS therapy after contracting the SARS-CoV-2 infection. Further research is required to delve into this matter.“

Overall, this scientific paper is well-structured and provides a thorough investigation into the impact of pre-existing ICS use on COVID-19 outcomes. It adheres to established guidelines and offers valuable insights into a relevant clinical question.

Reply: Thank you for your comment.

Reviewer #2: Dear Authors,

The research topic, which examines the effects of previous use of inhaled corticosteroids on the outcomes of COVID-19, is not a new subject. However, the systematic review and meta-analysis has been conducted with acceptable precision and well documented. The only recommendation is to include the number of studies used in each subgroup analysis within the text.

Reply: Thank you for your comment. We have included a new column titled “number of studies” in Table 2 to clearly indicate the number of studies involved in each subgroup analysis for each outcome.

---

## [Decision Letter · Decision Letter 1]

21 Nov 2023

The Influence of Prior Use of Inhaled Corticosteroids on COVID-19 Outcomes: A Systematic Review and Meta-Analysis

PONE-D-23-28841R1

Dear Dr. Fu,

We’re pleased to inform you that your manuscript has been judged scientifically suitable for publication and will be formally accepted for publication once it meets all outstanding technical requirements.

Kind regards,

Dong Keon Yon, MD, FACAAI, FAAAAI

Academic Editor

PLOS ONE

Additional Editor Comments (optional):

This is an excellent paper!

Reviewers' comments:

Reviewer's Responses to Questions

**Comments to the Author**

1. If the authors have adequately addressed your comments raised in a previous round of review and you feel that this manuscript is now acceptable for publication, you may indicate that here to bypass the “Comments to the Author” section, enter your conflict of interest statement in the “Confidential to Editor” section, and submit your "Accept" recommendation.

Reviewer #1: All comments have been addressed

2. Is the manuscript technically sound, and do the data support the conclusions?

Reviewer #1: Yes

3. Has the statistical analysis been performed appropriately and rigorously? 

Reviewer #1: N/A

4. Have the authors made all data underlying the findings in their manuscript fully available?

Reviewer #1: Yes

5. Is the manuscript presented in an intelligible fashion and written in standard English?

Reviewer #1: Yes

6. Review Comments to the Author

Reviewer #1: You have made appropriate changes to the manuscript. The manuscript is technically sound, and the data support the conclusions. I cannot give analysis to the statistical process .

7. PLOS authors have the option to publish the peer review history of their article (what does this mean?). If published, this will include your full peer review and any attached files.

Reviewer #1: No

---

## [Editor Report · Acceptance letter]

9 Jan 2024

PONE-D-23-28841R1 

PLOS ONE

Dear Dr. Fu, 

I'm pleased to inform you that your manuscript has been deemed suitable for publication in PLOS ONE. Congratulations! Your manuscript is now being handed over to our production team.

Kind regards, 

on behalf of

Dr. Dong Keon Yon 

Academic Editor

PLOS ONE